# Predictors for Uptake of Vaccines Offered during the Second Year of Life: Second Dose of Measles-Containing Vaccine and Meningococcal Serogroup A-Containing Vaccine, Ghana, 2020

**DOI:** 10.3390/vaccines11101515

**Published:** 2023-09-23

**Authors:** Pierre Muhoza, Monica P. Shah, Hongjiang Gao, Kwame Amponsa-Achiano, Pamela Quaye, William Opare, Charlotte Okae, Philip-Neri Aboyinga, Kwadwo L. Opare, Melissa T. Wardle, Aaron S. Wallace

**Affiliations:** 1Global Immunization Division, Center for Global Health, Centers for Disease Control and Prevention, Atlanta, GA 30329, USA; 2Epidemic Intelligence Service, Centers for Disease Control and Prevention, Atlanta, GA 30329, USA; 3Expanded Programme on Immunisation, Disease Control and Prevention Department, Public Health Division, Ghana Health Service, Accra 00233, Ghana; 4Neglected Tropical Diseases Control Programme, Disease Control and Prevention Department, Public Health Division, Ghana Health Service, Accra 00233, Ghana

**Keywords:** immunization, measles-containing vaccine dose, meningococcal serogroup a conjugate vaccine, life course, immunization schedule, immunization disparities

## Abstract

Background: Understanding the drivers of coverage for vaccines offered in the second year of life (2YL) is a critical focus area for Ghana’s life course approach to vaccination. This study characterizes the predictors of vaccine receipt for 2YL vaccines—meningococcal serogroup A conjugate vaccine (MACV) and the second dose of measles-containing vaccine (MCV2)—in Ghana. Methods: 1522 children aged 18–35 months were randomly sampled through household surveys in the Greater Accra Region (GAR), Northern Region (NR), and Volta Region (VR). The association between predictors and vaccination status was modeled using logistic regression with backwards elimination procedures. Predictors included child, caregiver, and household characteristics. Results: Coverage was high for infant vaccines (>85%) but lower for 2YL vaccines (ranging from 60.2% for MACV in GAR to 82.8% for MCV2 in VR). Predictors of vaccination status varied by region. Generally, older, first-born children, those living in rural settlements and those who received their recommended infant vaccines by their first birthday were the most likely to have received 2YL vaccines. Uptake was higher among those with older mothers and children whose caregivers were aware of the vaccination schedule. Conclusions: Improving infant immunization uptake through increased community awareness and targeted strategies, such as parental reminders about vaccination visits, may improve 2YL vaccination coverage.

## 1. Introduction

Immunization prevents numerous childhood infectious diseases, including measles and meningitis, both of which are highly contagious and have the potential for serious sequelae. Although substantial progress in measles control has been achieved through routine immunization (RI) services and supplementary immunization activities (SIAs), measles continues to be a significant cause of mortality, particularly in sub-Saharan Africa, where an estimated 4.4 million cases and 66,000 deaths in 2021 were attributed to measles [1]. A coverage level of 95% or higher with two doses of measles-containing vaccine (MCV) is needed to create herd immunity to protect communities and advance toward measles elimination; at 41% coverage, the World Health Organization (WHO) African Region is currently short of the goal [1]. Likewise, bacterial meningitis remains an enormous strain on the public health system across the region with an estimated mean annual incidence of 216 cases per 100,000 inhabitants [2]. Before the introduction of meningococcal serogroup A conjugate vaccine (MACV) in 2010 through mass vaccination campaigns, *Neisseria meningitidis* serogroup A accounted for 80–85% of meningitis epidemics in the “African meningitis belt” stretching from Senegal to Ethiopia.

The WHO recommends the routine administration of MACV and the second dose of the measles-containing vaccine (MCV2) during the second year of life (2YL) [3,4]. Ghana completed an initial MACV mass campaign in 2012 and conducted a catch-up campaign for children aged 1–4 years in July 2016 [5]. In November 2016, Ghana was the second country in the meningitis belt, after Sudan, to introduce MACV into the routine Expanded Programme on Immunization (EPI) schedule. MCV2 was introduced in the Ghana EPI schedule as a measles–rubella combination vaccine in February 2012. MCV2 and MACV are scheduled to be administered at the 18-month well-child visit.

Although Ghana has sustained national coverage rates of at least 85% for all infant vaccinations (those given before 12 months of age) since 2008, challenges have hampered similar success for vaccinations offered during 2YL [6]. A 2016 study, estimating vaccination coverage levels in the first and second year of life across three regions in Ghana before MACV introduction, revealed that infant vaccination coverage levels, including the MCV1 dose, were high (≥87%). However, coverage for MCV2 ranged from 60% to 83% across the study regions with high dropout rates between MCV1 and MCV2 [7].

Previous studies have highlighted a range of factors that influence 2YL vaccine uptake. Such factors include child and caregiver socio-demographic characteristics, caregiver knowledge and attitudes towards 2YL vaccinations, utilization of maternal care services as well as practical constraints related to bringing older children for 2YL vaccination visits and other competing priorities for limited caregiver time [7,8,9,10,11]. Health system factors related to the availability and quality of immunization services including distance to health facilities, vaccine stockouts, and caregiver perception of quality of care during immunization visits have also been cited as important for 2YL vaccine uptake [8,10,11,12,13]. However, previous studies have either been qualitative or descriptive in nature [7,8,9,10], limited in geographic scope [9,12,14,15] and largely focused on MCV2.

To increase coverage for 2YL vaccines in Ghana and plan for improved health services utilization, decision makers need to have an updated understanding of the determinants of 2YL immunization status in the country. As a follow-up to the 2016 study which was descriptive and predated MACV introduction in Ghana, we assessed predictors of MCV2 and MACV vaccination among children aged 18–35 months across three regions of Ghana in 2020.

## 2. Methods

### 2.1. Study Sites and Survey Design

Study data were derived from a cross-sectional endline survey conducted in August 2020 to estimate vaccination coverage among children aged 12–35 months old as part of a broader project to strengthen the 2YL RI platform in Ghana [7]. We used a stratified multi-stage cluster sample design with sampling stratified by three regions (Greater Accra [GAR], Northern [NR], and Volta [VR] Regions shown in Figure 1). In the first stage, 141 enumeration areas (EA) (53 in GAR, 52 in NR, and 36 in VR) were selected by probability proportional to size (PPS). Within each selected EA, a household listing was conducted to determine the sampling frame of households with at least one child aged 12–35 months as of March 2020. The selection of the cutoff was intended to minimize potential confounding related to the disruptions caused by the COVID-19 pandemic [16]. In the second stage, households with at least one age-eligible child were selected by simple random sampling. If a household had more than one age-eligible child, one child was selected at random. A standardized questionnaire was administered by the survey teams to the selected child’s caregiver who was present at the household during the day of the survey. Survey questions included child, caregiver, household characteristics and demographics; caregiver’s immunization awareness, knowledge, attitudes, beliefs and perceptions regarding family, community, and healthcare worker support for immunizations; and childhood vaccination history assessed by home-based vaccination record (child health record booklet-CHRB) data, health facility records, or caregiver recall.

Sample sizes were calculated to measure a difference of 15% in MCV2 coverage among children 24–35 months of age from the baseline survey in 2016 to the post-intervention endline survey in 2020. We assumed 80% power at a significance level of alpha = 0.05 using a one-sided test and adjusted for a 10% non-response rate. The estimated proportions of households with eligible children were 7.3% in GAR, 18.0% in NR, and 9.4% in VR, respectively. The intra-class correlation coefficients were 0.18 in GAR, 0.50 in NR, and 0.30 in VR. Sample sizes were then calculated separately for each of the three regions, using results from the 2014 Ghana Demographic and Health Survey (GDHS) that provided region-level estimates for their indicators [17].

### 2.2. Study Population

Although data were collected for children aged between 12 and 35 months at the time of the endline survey, the present study was restricted to a subset of children aged 18–35 months, since this older age group was either due or had already passed the point of the recommended age at vaccination for MACV and MCV2 doses.

### 2.3. Variables

#### 2.3.1. Measured Variables

MACV and MCV2 vaccination status at the time of the interview were analyzed as separate outcomes and dichotomized as single versus zero doses of MACV or two doses versus one or zero doses of MCV, respectively. Vaccination status was ascertained preferentially from either (1) the CHRB, (2) the child’s immunization record from health facility registers, or (3) based on caregiver recall. Vaccination coverage was defined as the proportion of children aged 18–35 months receiving the recommended vaccines. MCV1–MCV2 dropout was defined as the proportion of children aged 18–35 months having received MCV1, but not MCV2.

Variables examined as potential predictors of vaccine receipt were broadly classified into three categories (child, caregiver, and household) and analyzed at the child level. In addition to the child’s vaccination history, additional child-level characteristics included variables with known association to vaccination status: age, sex, birth order, and availability of RI services at the child’s school or daycare [18,19,20]. 

We considered caregiver variables that are known to be associated with their children’s immunization status, including maternal age at the time of the interview, highest educational level completed, occupation, and religion [21,22,23,24]. The survey included questions about caregivers’ awareness of vaccine schedules, attitudes toward vaccines in general (including attitudes about the importance, safety, and effectiveness of vaccines), 2YL vaccines specifically, vaccine-preventable infection and disease, source of vaccine-related information, perception of support for immunizations, and decision making on healthcare seeking and 2YL vaccination.

For household characteristics, we also included the settlement type (rural or urban residence). The full list of variables assessed as predictors in this study are shown in Appendix A.

#### 2.3.2. Derived Variables

We derived variables to examine whether MACV and MCV2 coverage differed by timely receipt of infant vaccines scheduled at 6 weeks, 14 weeks, and 9 months of age. A dose was considered timely if the vaccine was provided within 4 weeks after the recommended age of administration according to Ghana’s EPI schedule (Table 1). Delayed vaccinations were defined as any vaccines received beyond this interval. In defining timely administration of recommended vaccines, doses were coded as missing if they were implausible (i.e., dose before a child’s date of birth), too early (i.e., dose received before the recommended age), or if the date of receipt could not be verified from the CHRB or the facility immunization register. For each scheduled age-based vaccination visit (i.e., 6 weeks, 14 weeks, and 9 months), timely receipt was coded as the timely receipt of all recommended vaccines versus untimely (i.e., early or delayed) receipt of one or more doses. 

In univariate and multivariable analyses, we also examined whether the receipt of all infant vaccines by the age of one year was predictive of 2YL vaccine receipt. These sub-analyses excluded vaccination information derived from caregiver recall. Although the inactivated polio vaccine (IPV) was introduced to the national immunization schedule in June 2018 to be given to children aged 14 weeks, we excluded its receipt from analysis due to the potential time lag with scale-up and standardized recording of a newer vaccine. 

### 2.4. Statistical Analysis

Design-based descriptive analyses were used to examine proportions, tabulations, and summary measures (i.e., median, range) for summarizing binary, categorical, and continuous variables, respectively. The Chi-square test was used to compare proportions for categorical variables. The univariate association between each predictor and either MCV2 or MACV receipt was modeled separately, using binomial logistic regression. We used multivariable logistic regression to model the adjusted association between multiple predictors and each outcome (MACV or MCV2) separately. Backward elimination was used to select the final (reduced) model containing statistically significant predictors of vaccination receipt. As a first step of model reduction, the statistical significance between each potential predictor and MCV2 and MACV receipt separately was estimated, using simple logistic regression. All potential predictors with *p*-values ≤ 0.2 in the univariate analysis were retained in the initial full multivariable logistic regression models. We then used a stepwise backward elimination approach to systematically remove predictors. Variables statistically associated with MCV2 or MACV receipt (*p* < 0.05) remained in the final model. Due to established associations based on literature, several demographic characteristics of the child (age, sex, birth order), mother (age, education, religion), and household (urban/rural residence) were specified as fixed predictors and were included in all models irrespective of statistical significance [11,22,25,26]. We also assessed multicollinearity between predictor variables, using variance inflation factors (VIF), and considered VIFs above 2.5 as indicative of multicollinearity [27]. For each outcome, we conducted model-building analyses separately for each region. 

We considered the possibility of a child’s attendance at a school/daycare offering RI or caregiver awareness of the 18-month visit differentially affecting MCV2 or MACV immunization status depending on urban/rural residence and tested separately for significant interactions between these variables. 

All analyses were performed, using the appropriate clustering and survey weighting statements to account for the complex survey design described above. The Taylor series linearization method was used to calculate the variance of the parameter estimates. We used Stata version 17 for all data analysis [28]. Unadjusted and adjusted odds ratios (ORs) and 95% confidence intervals (CIs) were estimated from model output parameters.

## 3. Results

### 3.1. Sample Characteristics

We collected data for 1990 children aged 12–35 months from 1712 households across the three regions and included data for 1522 children (76.5%) aged 18–35 months for the analysis of determinants of 2YL vaccination status. The median age of these children was 26 months (interquartile range [IQR]: 22–31 months), and 50.8% were males. The child’s mother was the primary respondent for 95.8% of the children in GAR, 92.8% in NR, and 90.7% in VR. The median maternal age was 30 years (IQR: 25–35 years). In GAR, 87.6% of the children lived in urban settlements whereas in NR and VR, 73.2% and 70.5%, respectively, lived in rural areas. Half of the children (51.2%) were third or higher in birth order. In NR, most mothers (70.5%) were Muslim, while in GAR and VR the majority (84.9%; 86.3%) were Christian. Most mothers in GAR (83.2%) and VR (63.9%) attained secondary education or higher compared to only 20.7% in NR. Nearly all children (96.9%) had their immunization history verified from CHRBs or health facility registers (Table 2 and Appendix A). 

### 3.2. Vaccination Coverage

Across all regions, coverage was higher for infant vaccines compared to 2YL vaccines; and higher for MCV2 as compared with MACV (Table 1). Coverage generally decreased with each successive dose in the series as the recommended age of administration increased. Among those with documented vaccination dates, the proportion of children fully immunized with infant vaccines plus 2YL vaccines ranged from 50.1% (95% CI, 44.2–56.0) in NR to 62.3% (95% CI, 53.1–71.4) in VR. For all individual vaccine doses, coverage rates were lowest in GAR with MACV coverage at 60.2% (95% CI, 53.3–66.8) and MCV2 coverage at 67.4% (95% CI, 62.0–72.3). Coverage was highest for all doses in VR with MACV coverage at 77.7% (95% CI, 70.4–83.6) and MCV2 coverage at 82.8% (95% CI, 76.0–87.9). In NR, MACV coverage was 62.2% (95% CI, 56.3–67.7) whereas MCV2 coverage was 69.8% (95% CI, 63.2–75.8). 

Among the 1475 children with documented immunization history, 986 (66.8%) had vaccination service dates available for both MACV and MCV2; 6 (0.4%) children lacked service date documentation for both 2YL vaccines, 129 (8.7%) received only one of the two vaccines, and 354 (24.0%) missed both. Among those (*n* = 986) who received both vaccines, 859 (87.1%) received them on the same date and 127 (12.9%) received them on different dates. Of the 129 who received only one of the 2YL vaccines, 118 (91.5%) were vaccinated for MCV2 only, and 74 (57.4%) of these instances occurred during 2019. The MCV1 to MCV2 dropout rate ranged from 12.6% (95% CI, 7.9–17.3) in VR to 23.8% (95% CI, 18.4–29.3) in NR.

MACV and MCV2 vaccination coverage by various characteristics are respectively shown in Table 3 (A and B) stratified by region. Across all regions, coverage for both vaccine doses was high among children attending the 18-month well-child visit, ranging from 67.7% (95% CI [59.8–74.6]) for MACV in GAR to 85.2% (95% CI [78.1–90.4]) for MCV2 in VR (Table 3 (A and B)). MACV and MCV2 coverage was higher among children who were fully immunized by 12 months of age, although the difference was not statistically significant for either dose in VR. Furthermore, children who received all vaccine doses recommended for the 9-month scheduled visit generally had higher MACV and MCV2 coverage rates as compared with children who had at least one delayed dose during the visit. The same trend did not hold for vaccine doses recommended during the 6 week and 14 week of age visits (Table 3 (A and B)).

### 3.3. Predictors of MACV and MCV2 Vaccine Receipt

The unadjusted and adjusted (final, reduced model) associations between child, caregiver, and household factors and MACV and MCV2 receipt are shown in Table 4, respectively. 

#### 3.3.1. Child-Level Predictors

In all regions, children aged 24–35 months were more likely to be vaccinated with 2YL vaccines compared with those aged 18–23 months with aORs ranging from 3.0 (95% CI, 2.1–4.2) for MCV2 in NR to 7.2 (95% CI, 3.3–15.8) for MACV in VR. Higher birth order (third or higher) was significantly associated with lower odds of vaccination for MCV2 compared to being born first in both GAR (aOR: 0.4; 95% CI, 0.2–0.9) and VR (aOR: 0.5; 95% CI, 0.3–0.8), and for MACV in VR only (aOR: 0.4; 95% CI, 0.2–0.7). Being fully immunized with all infant vaccine doses by 12 months of age was positively associated with both MACV and MCV2 receipt in GAR and NR, but not VR. 

#### 3.3.2. Caregiver-Level Predictors

Children of older mothers were more likely to be vaccinated with either MACV or MCV2; this association was significant in GAR and NR but not in VR. In NR only, children of Christian mothers were twice as likely to receive MACV (aOR: 1.9, 95% CI [1.1–3.5]). There was no statistically significant association between caregiver education and occupation and MCV2 or MACV in any region.

Factors related to caregiver knowledge, attitudes, and beliefs around 2YL vaccination were found to exert some influence on 2YL vaccination status. In GAR, children of caregivers who were aware that MCV was recommended during the second year of life were almost twice as likely to receive MCV2 compared to those who were unaware (aOR:1.8; 95% CI, 1.1–2.9). In both NR and VR, children of caregivers who were aware of the 18-month child well-child visit were more likely to receive either 2YL vaccine with aORs ranging from 1.7 (95% CI, 1.16–2.6) for MCV2 in NR to 2.8 (95% CI, 1.7–4.5) for MACV in VR. In NR only, children of caregivers who believed in the importance of vaccination during the second year of life were more likely to be vaccinated with either MACV (aOR: 1.9; 95% CI, 1.1–3.4) or MCV2 (aOR: 2.9; 95% CI, 1.4–5.7). 

#### 3.3.3. Household-Level Predictors

Compared to urban areas, children living in rural areas were more likely to be vaccinated for MCV2 and MACV; however, the association was statistically significant only in GAR for MACV (aOR: 2.7; 95% CI, 1.2–5.9) and in NR for MACV (aOR:1.9; 95% CI, 1.2–3.0) and MCV2 (aOR: 2.6; 95% CI, 1.6–4.4). 

#### 3.3.4. Interaction Analysis

Although initial interaction analysis suggested that the effect of urban/rural residence in GAR depended on the availability of RI services at the child’s school (interaction term *p* value = 0.03), further assessment suggested that strata-specific effects were not statistically significant (Appendix A). Therefore, the interaction term was dropped from the final model for GAR. Nonetheless, among children in urban areas of GAR, the proportion of children vaccinated with MCV2 among those attending schools with RI services was 76.9% (95% CI, 67.5–84.3) compared to 62.6% (95% CI, 55.9–68.9]) among those attending schools without the services (Chi-square *p* value = 0.01). In rural areas of GAR, however, the MCV2 coverage was not significantly different among children attending schools with and without RI services (Appendix A).

## 4. Discussion

The present study found low uptake of MACV and MCV2 among children aged 18–35 months living in three regions of Ghana in 2020 and identified several predictors of vaccine receipt during the second year of life. Vaccination coverage for 2YL vaccines was lower than the infant vaccination coverage levels and below the goal of “95% coverage for all antigens by 2019” set forth by the Ghanaian Government [29]. MCV1 to MCV2 dropout rates were substantial (12.6% to 23.8%) suggesting further opportunity to strengthen the 2YL vaccination platform. Our results indicate that predictors of 2YL vaccination include characteristics for the child (age, birth order, and immunization history), caregiver (awareness of the RI schedule, attitudes about the importance of 2YL vaccination, and maternal age) and the household (urban–rural residence). 

Consistent with prior literature on childhood immunization, older children across all regions were more likely to be vaccinated with either dose compared to younger children. Although older children have a comparatively longer window of opportunity to receive the vaccine, it is also possible that the finding could reflect delays in vaccination, as studies from other low-and middle-income countries have suggested [30,31]. Further research on the timing or age of 2YL vaccine receipt could provide a better understanding of the latter hypothesis.

Increasing birth order generally reduced the likelihood of being vaccinated with 2YL vaccines. This was in line with previous research on infant immunizations in Ghana and elsewhere in Africa [22,32]. Caring for multiple children places competing demands on caregivers, with the time and resources required to meet an individual child’s healthcare needs decreasing with each additional child. Furthermore, given the importance of family support for childhood vaccination [33], those without it may experience even greater barriers. In NR, factors related to caregiver perception of spousal and other family support for 2YL immunization were significant in univariate analysis but were dropped in the final model. Possibly, the effect of these factors may be masked by other more influential factors identified in the multivariable analysis. Additional research is needed to better understand how household decision-making processes and social support for immunizations influence 2YL vaccination, particularly among large families. Nonetheless, it is concerning that younger children from larger families are less likely to receive the protection of MACV and MCV2 since these children are also more likely to be exposed to meningitis and measles by older siblings and other household members [34,35]. Leveraging health visits by caregivers from large families and home visits by healthcare workers to such families to provide tailored public health messaging on the importance of all vaccinations, including 2YL doses for younger children should be considered.

Receipt of all recommended infant vaccines prior to one year of age was strongly predictive of 2YL vaccine receipt in both GAR and NR. This finding highlights that early immunization adherence can lead to continued immunization uptake throughout childhood; 2YL vaccine coverage could be promoted by leveraging health visits during the first year of life, which are crucial opportunities to educate caregivers about the benefits of vaccination and the need to complete their child’s vaccination schedule. We found that 2YL vaccine coverage was higher among children receiving timely 9-month visit vaccinations, but not for children receiving timely 6- or 14-week visit vaccinations, which suggests that sustained vaccine schedule compliance during the first year of life may improve 2YL vaccination uptake. Possible interventions to improve 2YL vaccine uptake include strengthening the tracking of children who receive vaccinations in the first year of life, but fail to receive vaccinations in the second year of life (i.e., defaulters), improving the health system’s ability to consistently reach caregivers of under-vaccinated children, and employing simple-to-use reminder systems to improve caregiver adherence to the immunization schedule. 

Caregivers’ awareness of the RI schedule was an important determinant of childhood vaccination uptake in our study. Children of caregivers who were aware of the 18-month well-child visit were significantly more likely to be vaccinated with 2YL vaccines in NR and VR. Similarly, in GAR, children of caregivers who knew that an MCV dose is needed before the second birthday were more than twice as likely to receive MCV2. These findings are consistent with a recent qualitative assessment of behavioral drivers of MACV and MCV2 vaccination among caregivers in Burkina Faso, which also concluded that knowledge gaps about the 2YL vaccination visit were a key barrier to vaccine uptake [8]. Improving awareness of the 18-month vaccination visit may increase attendance at these visits and improve 2YL vaccine coverage. Across all regions in our study, at least three-quarters of the children attending the 18-month visit received MCV2 and at least two-thirds received MACV. Altogether, these findings emphasize the need to promote the 18-month visit which can be accomplished by providing reminders to caregivers and targeted communication to communities. Ensuring that healthcare providers and community health workers receive adequate training and support in interpersonal communication regarding vaccination schedules, particularly during the second year of life, may also improve uptake. The interpersonal communication training will also need to emphasize respectful treatment of clients given previous reports that caregivers who have missed scheduled visits or bring older children for 2YL vaccinations sometimes experience rude or condescending treatment from healthcare providers [8,10]. 

Additionally, our findings highlight the critical role of caregiver attitudes in improving childhood vaccination [36,37,38]. Of the behavioral factors we examined, belief in the importance of 2YL vaccination emerged as an important predictor of a child’s vaccination status with both MACV and MCV2 among caregivers in NR. This finding is consistent with a previous assessment that found caregiver perception of the combined benefits and risks of vaccination was the strongest predictor of vaccine compliance relative to the other factors evaluated in this context [36]. Accordingly, public health messaging in northern Ghana could be further tailored to address the concerns about vaccine safety and to communicate the benefits of timely vaccination, particularly during the second year of life. More broadly, demand generation activities should be tailored to address caregiver demographic and cultural differences across regions. For instance, different communication approaches could be considered in NR where most caregivers are Muslim and with low literacy levels compared with those in GAR who are mostly Christian and with higher education levels. In areas such as NR where religious affiliation appears to influence 2YL vaccination, it is possible that greater involvement of religious leaders and faith-based organizations in vaccine promotion may be an effective strategy for building community trust and increasing vaccine confidence which in turn may increase uptake [39,40,41]. 

Children of mothers aged 25 years or older were more likely to be vaccinated with either MACV or MCV2 as compared to children of younger mothers. Although the associations were not significant across all regions, the directionality of estimates was consistent across models. Previous research assessing the effect of maternal age on child immunization status in Ghana has focused on infant vaccines with some studies finding positive, negative, or inconclusive associations [26,32,42]. Consistent with our study, a recent secondary analysis from eight sub-Saharan African countries with publicly available MCV2 data found that older maternal age was associated with a higher likelihood of vaccination with MCV2 [11]. Compared with younger mothers, older mothers may have more resources at their disposal and more bargaining power with other family members in intra-household decision-making processes related to their child’s health [43,44]. It is also possible that older mothers have higher levels of immunization-related knowledge than younger mothers [45,46]. 

Similar to previous studies focused on infant vaccination in Ghana, we noted urban–rural disparities with children from rural areas being more likely to be vaccinated with 2YL vaccines [25,32]. This was particularly evident for both MACV and MCV2 in NR, and for MCV2 only in GAR where residence in rural settlements was highly predictive of vaccine receipt. The rural advantage has been observed for infant vaccines since at least 2014 and is likely explained by a combination of the unequal success of Ghana’s Community-based Health Planning and Service (CHPS) initiative that had implementation challenges in urban areas, and the challenges with primary healthcare service delivery in rapidly growing informal settlements in urban areas [25,32,47]. Future immunization program planning will need to focus on underserved urban communities to address the persistent immunization inequities, particularly in GAR, which had the lowest 2YL vaccine coverage rates and highest urban population. Our findings also suggested that service delivery in urban areas of GAR could be augmented by school-based vaccination programs to increase access and close coverage gaps. Nonetheless, additional research is needed to further clarify the effectiveness of intensifying the integration of 2YL vaccination into school-based health services and daycares in urban areas.

Across all regions, coverage with MACV was lower than MCV2 coverage. This could be due to the lag effects of the more recently introduced MACV on coverage or national-level MACV stockouts for two months in 2019 due to funding delays [48]. This is consistent with our observation that nearly half of the vaccination visits MACV and MCV2 were not co-administered occurred in 2019, ultimately resulting in missed opportunities for MACV vaccination. Although children in both age groups in our study were impacted by the MACV stockouts, it is difficult to determine to what extent the stockouts impacted the observed associations. Of note, Ghana experienced disruptions to immunization services during 2020 due to the COVID-19 pandemic [49]. Since we only analyzed immunizations received prior to the beginning of the pandemic, we are unable to examine the extent to which 2YL vaccine coverage was affected by the disruptions.

## 5. Strengths and limitations

Our study has several strengths. First, it provides important insights into the predictors of vaccination during 2YL in Ghana, which was among the first countries in the African region to introduce 2YL vaccines [5,13]. Second, the data used for these analyses come from a household vaccination coverage survey specifically tailored to assess barriers to 2YL vaccine uptake. Third, survey methodology was used to account for the survey sampling design which minimized bias in the calculated estimates. Finally, vaccination status was verified using CHRBs and facility registers for 96.9% of the children in the study, thereby reducing the possibility of outcome misclassification bias. 

This study is also subject to some limitations. First, we were unable to distinguish routine MCV doses from supplemental doses delivered through the MCV SIA campaign in Ghana during 2018 [50]. Since the survey was conducted more than a year after completion of the SIA campaign and because supplemental doses given are typically not designated as such on the CHRB, there could have been recall bias in ascertainment of MCV vaccination status. Additionally, many variables were measured based on respondent self-report, which may be subject to recall bias. Furthermore, this survey is cross-sectional so we cannot establish a temporal relationship between the examined predictors and 2YL vaccine receipt. We can only make claims on statistical associations; hence no causal inference can be made. Finally, there may be important predictors of vaccination receipt that were unmeasured in this study. For instance, place of delivery, socioeconomic status, accessibility, and convenience of vaccination services are known to influence immunization outcomes, but these data were not collected as part of the household survey. Wherever possible, we used proxy variables that could capture the effect of the omitted variable. For instance, the inclusion in our models of caregiver occupation and education levels may also capture caregiver socioeconomic status in Ghana as previously done [25]. We also did not assess the impact on 2YL vaccine uptake for health facility factors such as healthcare worker knowledge and attitudes towards immunization, adherence to EPI guidelines, workload, or vaccination supply stock practices. Future research should incorporate these aspects as they likely influence 2YL vaccine uptake.

## 6. Conclusions

Using data from three regions of Ghana, we found that MACV and MCV2 vaccination uptake remains lower than the infant vaccination uptake. Thus, there is room for improvement in reaching the Ghana government’s vaccination targets. Predictors of vaccination during 2YL varied across regions and included sociodemographic characteristics, as well as caregiver knowledge, attitudes, and beliefs toward vaccination. Disparities persist with children in urban areas being less likely to be vaccinated. Effective interventions will require multifaceted and context-specific approaches to improve immunization compliance overall while specifically promoting the 18-month well-child visit. Robust tracking of defaulters, targeted communication to communities, and reminders to caregivers may improve 2YL vaccine coverage. Caregivers, particularly younger mothers, with multiple children likely experience practical barriers to bringing their children for 2YL vaccination. Leveraging every health facility or home visit contacts with such caregivers to emphasize the importance of immunizations and schedule adherence should be a key consideration to increase 2YL vaccine uptake. 

## Figures and Tables

**Figure 1 vaccines-11-01515-f001:**
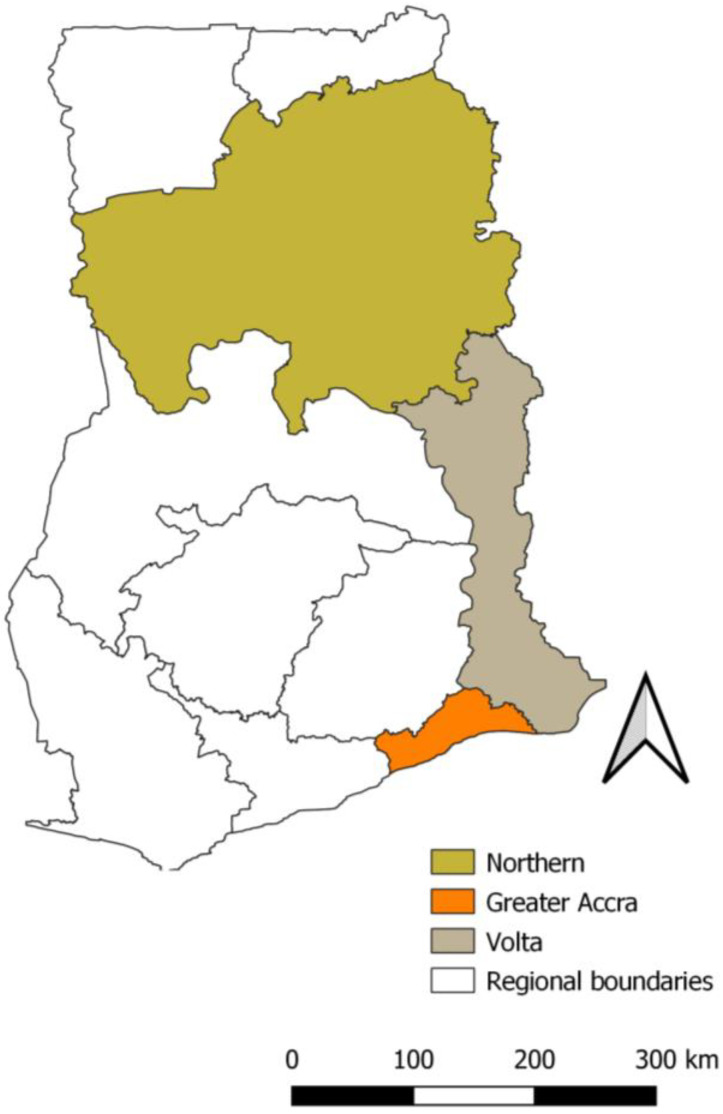
Map of Ghana showing the location of the Greater Accra, Northern, and Volta regions based on administrative boundaries from 2018.

**Table 1 vaccines-11-01515-t001:** Ghana vaccine schedule, 2020, and antigen-specific vaccine coverage among children aged 18–35 months in Greater Accra Region (GAR), Northern Region (NR), and Volta Region (VR).

	Recommended Age	Vaccine Uptake % (95% CIs)		
Vaccine		GAR (*N* = 370)	NR (*N* = 870)	VR (*N* = 315)
BCG	At birth or first contact	81.3 (76.4–86.3)	93.8 (91.7–95.8)	87.9 (82.4–93.4)
OPV1	6 weeks	82.8 (77.9–87.8)	96.0 (94.4–97.5)	96.9 (94.1–99.6)
PCV1	6 weeks	82.9 (78.0–87.9)	96.6 (95.3–98.0)	96.9 (94.1–99.6)
Pentavalent 1	6 weeks	82.9 (78.0–87.9)	96.6 (95.3–98.0)	96.9 (94.1–99.6)
Rotavirus 1	6 weeks	82.7 (77.8–87.6)	95.4 (93.9–96.8)	96.8 (94.0–99.6)
OPV2	10 weeks	82.4 (77.5–87.4)	92.3 (89.7–94.8)	96.6 (93.8–99.3)
PCV2	10 weeks	82.9 (78.0–87.9)	96.1 (94.4–97.8)	96.8 (94.0–99.6)
Pentavalent 2	10 weeks	82.9 (78.0–87.9)	96.3 (94.8–97.8)	96.8 (94.0–99.6)
Rotavirus 2	10 weeks	81.0 (76.0–86.0)	92.3 (89.9–94.7)	94.3 (91.5–97.0)
IPV *	14 weeks			
OPV3	14 weeks	78.0 (72.8–83.2)	82.1 (78.0–86.2)	90.8 (84.9–96.7)
PCV3	14 weeks	82.1 (76.6–87.6)	93.9 (91.7–96.1)	95.7 (92.9–98.6)
Pentavalent 3	14 weeks	82.3 (76.8–87.8)	93.5 (91.3–95.6)	95.2 (92.3–98.0)
MCV1	9 months	81.0 (75.9–86.1)	91.7 (88.8–94.5)	94.7 (91.2–98.1)
Yellow Fever	9 months	81.2 (76.1–86.4)	88.2 (85.0–91.5)	93.6 (90.2–97.0)
MACV	18 months	60.2 (53.3–67.2)	62.2 (56.3–68.1)	77.7 (70.8–84.5)
MCV2	18 months	67.4 (62.1–72.6)	69.8 (63.3–76.3)	82.8 (76.6–88.9)
Fully immunized with infant vaccines **		80.6 (75.3–85.9)	71.7 (66.6–76.8)	77.9 (70.9–84.8)
Fully immunized with infant vaccines plus MCV2 **		67.1 (61.6–72.6)	56.4 (49.6–63.1)	67.1 (58.9–75.3)
Fully immunized with infant vaccines plus MCV2 & MACV **		62.8 (56.5–69.2)	50.1 (44.2–56.0)	62.1 (53.0–71.2)
MCV1 to MCV2 dropout rate		16.9 (13.1–20.6)	23.8 (18.4–29.3)	12.6 (7.9–17.3)

BCG, bacillus Calmette–Guérin; MACV, meningitis serogroup A conjugate vaccine; MCV, measles-containing vaccine includes rubella vaccine, OPV, oral polio vaccine; pentavalent vaccines include diphtheria, pertussis, tetanus, Hemophilus influenzae type b, and hepatitis B vaccines; PCV, pneumococcal conjugate vaccine. * IPV, inactivated polio vaccine was introduced in Ghana in 2018 and was excluded from analyses in this study. ** Children with vaccination information from caregiver recall excluded from sub-analysis.

**Table 2 vaccines-11-01515-t002:** Characteristics of children 18–35 months of age in Greater Accra (GAR), Northern (NR), and Volta (VR) regions, 2020 *.

	Region							
	GAR		NR		VR		Total	
	No.	%	No.	%	No.	%	No.	%
**Child’s age group (months)**								
18–23	146	39.5	302	36.1	115	36.5	563	37.0
24–35	224	60.5	535	63.9	200	63.5	959	63.0
Total	370	100.0	837	100.0	315	100.0	1522	100.0
**Child’s sex**								
Female	184	49.7	396	47.3	177	56.2	757	49.7
Male	186	50.3	441	52.7	138	43.8	765	50.3
Total	370	100.0	837	100.0	315	100.0	1522	100.0
**Birth order**								
First	122	33.0	182	21.7	101	32.1	405	26.6
Second	100	27.0	175	20.9	62	19.7	337	22.1
Third or more	148	40.0	480	57.3	152	48.3	780	51.2
Total	370	100.0	837	100.0	315	100.0	1522	100.0
**Maternal age (years)**								
<25	42	11.4	108	12.9	71	22.5	221	14.5
25–34	184	49.7	215	25.7	134	42.5	533	35.0
≥35	144	38.9	514	61.4	110	34.9	768	50.5
Total	370	100.0	837	100.0	315	100.0	1522	100.0
**Maternal education**								
Never attended school	25	6.8	592	70.9	52	16.6	669	44.1
Primary	37	10.0	70	8.4	61	19.5	168	11.1
Secondary or higher	308	83.2	173	20.7	200	63.9	681	44.9
Total	370	100.0	835	100.0	313	100.0	1518	100.0
**Paternal education**								
Never attended school	12	3.5	582	71.1	38	13.1	632	43.5
Primary	12	3.5	42	5.1	29	10.0	83	5.7
Secondary or higher	319	93.0	195	23.8	224	77.0	738	50.8
Total	343	100.0	819	100.0	291	100.0	1453	100.0
**Maternal religion**								
Christian	314	84.9	178	21.3	272	86.3	764	50.2
Muslim	55	14.9	590	70.5	23	7.3	668	43.9
Traditionalist	0	0.0	55	6.6	6	1.9	61	4.0
None/Don’t know	1	0.3	14	1.7	14	4.4	29	1.9
Total	370	100.0	837	100.0	315	100.0	1522	100.0
**Maternal marital status**								
Single/Divorced/Separated/Widowed	48	13.0	16	1.9	39	12.4	103	6.8
Married/Cohabitating	322	87.0	821	98.1	276	87.6	1419	93.2
Total	370	100.0	837	100.0	315	100.0	1522	100.0
**Settlement type**								
Urban	324	87.6	224	26.8	93	29.5	641	42.1
Rural	46	12.4	613	73.2	222	70.5	881	57.9
Total	370	100.0	837	100.0	315	100.0	1522	100.0
**Attends school/daycare offering routine immunization (RI) ****								
No formal school/daycare attendance	179	48.4	737	88.1	221	70.2	1137	74.7
School/daycare does not offer RI	90	24.3	81	9.7	48	15.2	219	14.4
School/daycare offers RI	101	27.3	19	2.3	46	14.6	166	10.9
Total	370	100.0	837	100.0	315	100.0	1522	100.0
**Timely receipt of all 6-week infant vaccines**								
At least 1 recommended dose delayed	40	13.7	282	38.1	53	18.7	375	28.5
All recommended doses timely	251	86.3	458	61.9	230	81.3	939	71.5
Total	291	100.0	740	100.0	283	100.0	1314	100.0
**Timely receipt of all 14-week infant vaccines**								
At least 1 recommended dose delayed	94	31.9	511	69.0	107	38.6	712	54.2
All recommended doses timely	201	68.1	230	31.0	170	61.4	601	45.8
Total	295	100.0	741	100.0	277	100.0	1313	100.0
**Timely receipt of all 9 mos. infant vaccines**								
At least 1 recommended dose delayed	87	30.5	381	57.5	123	48.0	591	49.1
All recommended doses timely	198	69.5	282	42.5	133	52.0	613	50.9
Total	285	100.0	663	100.0	256	100.0	1204	100.0
**Fully immunized child by 12 mos.**								
No	54	17.1	349	43.2	89	29.2	492	34.5
Yes	261	82.9	458	56.8	216	70.8	935	65.5
Total	315	100.0	807	100.0	305	100.0	1427	100.0
**18 mos. well-child visit attendance**								
No	48	13.0	125	14.9	32	10.2	205	13.5
Yes	322	87.0	712	85.1	283	89.8	1317	86.5
Total	370	100.0	837	100.0	315	100.0	1522	100.0
**Caregiver aware of need for 18 mos. visit before 2nd birthday**								
No	163	44.1	452	54.0	151	47.9	766	50.3
Yes	207	55.9	385	46.0	164	52.1	756	49.7
Total	370	100.0	837	100.0	315	100.0	1522	100.0
**Caregiver knows MR dose should be given during 2YL**								
No	153	41.4	284	33.9	107	34.0	544	35.7
Yes	217	58.6	553	66.1	208	66.0	978	64.3
Total	370	100.0	837	100.0	315	100.0	1522	100.0
**Caregiver received phone reminder for 18 mos. visit**								
No access to phone	15	4.1	233	27.8	45	14.3	293	19.3
Access to phone but no reminder received	313	84.6	523	62.5	223	70.8	1059	69.6
Phone reminder received	42	11.4	81	9.7	47	14.9	170	11.2
Total	370	100.0	837	100.0	315	100.0	1522	100.0
**Caregiver believes healthcare provider provides enough information on 2YL vaccination**								
No	42	11.4	114	13.6	24	7.6	180	11.8
Yes	328	88.6	723	86.4	291	92.4	1342	88.2
Total	370	100.0	837	100.0	315	100.0	1522	100.0
**Caregiver believes 2YL vaccination is important for child**								
No	11	3.0	31	3.7	6	1.9	48	3.2
Yes	359	97.0	806	96.3	309	98.1	1474	96.8
Total	370	100.0	837	100.0	315	100.0	1522	100.0
**Caregiver believes 2YL vaccination is supported by spouse**								
No	22	5.9	29	3.5	21	6.7	72	4.7
Yes	348	94.1	808	96.5	294	93.3	1450	95.3
Total	370	100.0	837	100.0	315	100.0	1522	100.0
**Caregiver believes 2YL vaccination supported by other family members**								
No	34	9.2	53	6.3	17	5.4	104	6.8
Yes	336	90.8	784	93.7	298	94.6	1418	93.2
Total	370	100.0	837	100.0	315	100.0	1522	100.0
**Caregiver believes 2YL vaccination supported by friends**								
No	66	17.8	71	8.5	29	9.2	166	10.9
Yes	304	82.2	766	91.5	286	90.8	1356	89.1
Total	370	100.0	837	100.0	315	100.0	1522	100.0
**Caregiver believes 2YL vaccination supported by community**								
No	71	19.2	95	11.4	18	5.7	184	12.1
Yes	299	80.8	742	88.6	297	94.3	1338	87.9
Total	370	100.0	837	100.0	315	100.0	1522	100.0
**Most people caregiver knows bring child in for 2YL vaccination**								
No	42	11.4	118	14.1	21	6.7	181	11.9
Yes	328	88.6	719	85.9	294	93.3	1341	88.1
Total	370	100.0	837	100.0	315	100.0	1522	100.0
**Caregiver believes it is important for child to get missing doses**								
No	24	6.5	47	5.6	12	3.8	83	5.5
Yes	346	93.5	790	94.4	303	96.2	1439	94.5
Total	370	100.0	837	100.0	315	100.0	1522	100.0
**Caregiver informed of 18 mos. visit by healthcare provider**								
No	178	48.1	469	56.0	165	52.4	812	53.4
Yes	192	51.9	368	44.0	150	47.6	710	46.6
Total	370	100.0	837	100.0	315	100.0	1522	100.0
**Vaccination source for coverage estimates**								
Vaccination card	316	85.4	815	97.4	306	97.1	1437	94.4
Facility register	21	5.7	10	1.2	7	2.2	38	2.5
Caregiver recall	33	8.9	12	1.4	2	0.6	47	3.1
Total	370	100.0	837	100.0	315	100.0	1522	100.0
**Same day receipt of MCV2 and MACV *****								
No	34	15.1	70	13.4	23	9.6	127	12.9
Yes	191	84.9	452	86.6	216	90.4	859	87.1
Total	225	100.0	522	100.0	239	100.0	986	100.0

2YL, second year of life; MR, measles–rubella vaccine. * Results are unweighted. ** Information on school/daycare attendance refers to the pre-pandemic period. *** Among those having received both vaccines and with vaccination status for both verified using home-based immunization cards or facility records.

**Table 3 vaccines-11-01515-t003:** (**A**): MACV coverage among children aged 18–35 months in Greater Accra Region (GAR), Northern Region (NR), and Volta Region (VR), Ghana, 2020 *. (**B**): MCV2 coverage among children aged 18–35 months children in Greater Accra Region (GAR), Northern Region (NR), and Volta Region (VR), Ghana, 2020 *.

A									
		*GAR*			*NR*			*VR*	
	*n*	%	95% CI	*n*	%	95% CI	*n*	%	95% CI
*Child’s age group (months)*									
18–23	**146**	**48.9**	**(39.9–57.9)**	**302**	**48.7**	**(41.8–55.7)**	**115**	**57.8**	**(45.0–69.6)**
24–35	**224**	**67.8**	**(59.5–75.1)**	**535**	**70.1**	**(63.4–76.1)**	**200**	**90.0**	**(84.0–94.0)**
Total	**370**	**60.2**	**(53.3–66.8)**	**837**	**62.2**	**(56.3–67.7)**	**315**	**77.7**	**(70.4–83.6)**
*Child’s sex*									
Female	184	61.3	(52.0–69.8)	**396**	**66.6**	**(60.5–72.2)**	177	73.6	(62.7–82.2)
Male	186	59.2	(49.1–68.6)	**441**	**58.3**	**(50.6–65.6)**	138	83.5	(74.3–89.8)
Total	370	60.2	(53.3–66.8)	**837**	**62.2**	**(56.3–67.7)**	315	77.7	(70.4–83.6)
*Birth order*									
First	122	63.9	(53.1–73.4)	182	65.7	(55.7–74.5)	101	83.4	(74.4–89.7)
Second	100	58.5	(43.7–71.8)	175	60.9	(51.3–69.8)	62	83.3	(66.4–92.6)
Third or more	148	58.1	(48.8–66.8)	480	61.2	(55.3–66.8)	152	70.7	(59.7–79.7)
Total	370	60.2	(53.3–66.8)	837	62.2	(56.3–67.7)	315	77.7	(70.4–83.6)
*Maternal age (years)*									
<25	42	50.2	(32.2–68.2)	**108**	**51.0**	**(40.2–61.7)**	71	81.0	(66.3–90.2)
25–34	184	63.3	(54.0–71.8)	**215**	**68.3**	**(61.0–74.9)**	134	77.9	(70.2–84.1)
≥35	144	59.2	(49.2–68.5)	**514**	**62.1**	**(56.1–67.8)**	110	75.1	(64.4–83.5)
Total	370	60.2	(53.3–66.8)	**837**	**62.2**	**(56.3–67.7)**	315	77.7	(70.4–83.6)
*Maternal education*									
Never attended school	25	63.8	(45.2–79.1)	592	64.5	(58.8–69.8)	52	74.7	(64.5–82.8)
Primary	37	55.0	(37.8–71.1)	70	46.2	(29.5–63.8)	61	76.7	(61.7–87.1)
Secondary or higher	308	60.5	(52.8–67.6)	173	61.4	(51.6–70.3)	200	79.1	(69.4–86.4)
Total	370	60.2	(53.3–66.8)	835	62.1	(56.3–67.6)	313	78.0	(71.1–83.6)
*Paternal education*									
Never attended school	12	77.1	(56.0–89.9)	582	63.8	(57.3–69.8)	38	77.5	(60.2–88.7)
Primary	12	60.2	(28.2–85.3)	42	62.3	(40.7–79.9)	29	73.7	(51.9–87.9)
Secondary or higher	319	60.5	(53.4–67.1)	195	58.2	(49.6–66.3)	224	79.2	(70.1–86.0)
Total	343	60.8	(53.9–67.3)	819	62.3	(56.4–67.8)	291	78.5	(71.3–84.2)
*Maternal religion*									
Christian	314	60.9	(52.7–68.5)	**178**	**76.1**	**(67.1–83.3)**	272	78.4	(70.3–84.8)
Muslim	55	54.5	(40.2–68.1)	**590**	**58.1**	**(51.6–64.4)**	23	72.1	(56.1–83.9)
Traditionalist	0	0.0		**55**	**70.7**	**(59.0–80.2)**	6	76.7	(32.2–95.8)
None/Don’t know	1	0.0		**14**	**55.8**	**(34.3–75.3)**	14	72.0	(35.2–92.4)
Total	370	60.2	(53.3–66.8)	**837**	**62.2**	**(56.3–67.7)**	315	77.7	(70.4–83.6)
*Maternal marital status*									
Single/Divorced/Separated/Widowed	48	50.5	(34.0–66.9)	16	65.1	(44.5–81.3)	39	74.9	(50.6–89.7)
Married/Cohabitating	322	61.8	(54.8–68.3)	821	62.1	(56.2–67.7)	276	78.0	(71.6–83.4)
Total	370	60.2	(53.3–66.8)	837	62.2	(56.3–67.7)	315	77.7	(70.4–83.6)
*Settlement type*									
Urban	324	59.0	(51.8–65.8)	**224**	**52.5**	**(44.5–60.3)**	93	77.1	(63.2–86.8)
Rural	46	71.7	(59.8–81.2)	**613**	**66.2**	**(59.0–72.6)**	222	77.9	(68.9–84.9)
Total	370	60.2	(53.3–66.8)	**837**	**62.2**	**(56.3–67.7)**	315	77.7	(70.4–83.6)
*Attends school/daycare offering routine immunization (RI)*									
No formal school/daycare attendance	**179**	**51.6**	**(42.4–60.7)**	737	61.5	(55.0–67.5)	**221**	**73.5**	**(64.8–80.7)**
School/daycare does not offer RI	**90**	**62.6**	**(52.2–72.0)**	81	66.5	(55.4–76.0)	**48**	**95.6**	**(86.5–98.7)**
School/daycare offers RI	**101**	**72.9**	**(65.1–79.4)**	19	72.7	(50.9–87.3)	**46**	**81.0**	**(65.5–90.5)**
Total	**370**	**60.2**	**(53.3–66.8)**	837	62.2	(56.3–67.7)	**315**	**77.7**	**(70.4–83.6)**
*Timely receipt of all 6-week infant vaccines*									
At least 1 recommended dose delayed	40	82.7	(68.5–91.3)	282	62.9	(56.2–69.2)	53	83.5	(64.7–93.3)
All recommended doses timely	251	70.3	(63.8–76.0)	458	64.4	(57.7–70.6)	230	79.7	(73.0–85.0)
Total	291	71.9	(65.8–77.3)	740	63.9	(58.0–69.4)	283	80.3	(74.0–85.4)
*Timely receipt of all 14-week infant vaccines*									
At least 1 recommended dose delayed	94	73.4	(63.1–81.6)	511	64.2	(57.9–70.1)	107	81.8	(72.7–88.4)
All recommended doses timely	201	74.0	(66.6–80.3)	230	68.1	(61.2–74.2)	170	80.1	(72.4–86.0)
Total	295	73.8	(67.7–79.2)	741	65.5	(60.2–70.4)	277	80.7	(74.3–85.8)
*Timely receipt of all 9 mos. infant vaccines*									
At least 1 recommended dose delayed	87	72.2	(59.3–82.2)	381	66.3	(59.6–72.4)	123	80.9	(70.5–88.2)
All recommended doses timely	198	74.8	(66.8–81.5)	282	71.7	(64.7–77.9)	133	80.5	(73.3–86.1)
Total	285	74.1	(67.5–79.7)	663	68.7	(63.7–73.3)	256	80.7	(75.0–85.3)
*Fully immunized child by 12 mos.*									
No	**54**	**59.1**	**(44.3–72.4)**	**349**	**54.3**	**(46.9–61.6)**	89	77.9	(62.7–88.1)
Yes	**261**	**75.1**	**(68.8–80.4)**	**458**	**71.8**	**(66.6–76.4)**	216	80.8	(74.8–85.7)
Total	**315**	**72.5**	**(66.4–77.8)**	**807**	**64.3**	**(58.6–69.6)**	305	80.0	(74.0–84.9)
*18 mos. well-child visit attendance*									
No	**48**	**14.1**	**(5.4–32.0)**	**125**	**24.8**	**(14.4–39.3)**	**32**	**50.3**	**(29.2–71.3)**
Yes	**322**	**67.7**	**(59.8–74.6)**	**712**	**69.8**	**(65.0–74.2)**	**283**	**80.3**	**(72.6–86.2)**
Total	**370**	**60.2**	**(53.3–66.8)**	**837**	**62.2**	**(56.3–67.7)**	**315**	**77.7**	**(70.4–83.6)**
*Caregiver aware of need for 18 mos. visit before 2nd birthday*									
No	163	56.2	(46.7–65.3)	**452**	**55.1**	**(47.9–62.1)**	**151**	**68.5**	**(59.0–76.7)**
Yes	207	62.8	(52.6–72.1)	**385**	**70.9**	**(64.0–76.9)**	**164**	**85.8**	**(78.9–90.7)**
Total	370	60.2	(53.3–66.8)	**837**	**62.2**	**(56.3–67.7)**	**315**	**77.7**	**(70.4–83.6)**
*Caregiver knows MACV dose should be given during 2YL*									
No	153	56.2	(47.1–65.0)	284	59.4	(50.2–68.1)	107	73.7	(65.1–80.8)
Yes	217	63.4	(54.2–71.7)	553	63.7	(57.9–69.2)	208	80.3	(71.7–86.8)
Total	370	60.2	(53.3–66.8)	837	62.2	(56.3–67.7)	315	77.7	(70.4–83.6)
*Phone reminder for 18 mos. visit*									
No access to phone	15	30.6	(12.0–58.8)	233	66.0	(59.1–72.2)	45	65.7	(46.6–80.8)
Access to phone but no reminder received	313	60.2	(51.3–68.4)	523	59.8	(52.7–66.5)	223	78.5	(70.4–84.9)
Phone reminder received	42	69.5	(50.7–83.5)	81	70.1	(57.3–80.4)	47	84.6	(71.3–92.3)
Total	370	60.2	(53.3–66.8)	837	62.2	(56.3–67.7)	315	77.7	(70.4–83.6)
*Gets enough 2YL vaccination info from provider*									
No	42	62.3	(47.9–74.7)	114	54.7	(41.3–67.4)	24	78.1	(53.9–91.5)
Yes	328	60.0	(52.5–67.0)	723	63.3	(57.7–68.5)	291	77.6	(70.1–83.7)
Total	370	60.2	(53.3–66.8)	837	62.2	(56.3–67.7)	315	77.7	(70.4–83.6)
*Caregiver believes 2YL vaccine important for child*									
No	11	68.1	(43.4–85.5)	**31**	**43.5**	**(28.7–59.5)**	6	91.4	(54.0–99.0)
Yes	359	60.0	(52.9–66.8)	**806**	**63.1**	**(57.4–68.4)**	309	77.4	(69.9–83.5)
Total	370	60.2	(53.3–66.8)	**837**	**62.2**	**(56.3–67.7)**	315	77.7	(70.4–83.6)
*Caregiver believes 2YL vaccination supported by spouse*									
No	22	60.5	(38.1–79.3)	29	40.5	(19.9–65.1)	21	82.5	(65.0–92.3)
Yes	348	60.2	(53.2–66.9)	808	63.1	(57.5–68.3)	294	77.3	(69.6–83.5)
Total	370	60.2	(53.3–66.8)	837	62.2	(56.3–67.7)	315	77.7	(70.4–83.6)
*Caregiver believes 2YL vaccination supported by other family members*									
No	34	57.2	(41.9–71.3)	**53**	**40.3**	**(22.6–61.0)**	17	89.2	(67.3–97.0)
Yes	336	60.5	(53.5–67.2)	**784**	**63.7**	**(58.4–68.8)**	298	76.9	(69.1–83.2)
Total	370	60.2	(53. –-66.8)	**837**	**62.2**	**(56.3–67.7)**	315	77.7	(70.4–83.6)
*Caregiver believes 2YL vaccination supported by friends*									
No	66	54.8	(39.5–69.3)	71	55.3	(43.7–66.3)	29	90.0	(64.8–97.8)
Yes	304	61.4	(54.4–68.0)	766	62.9	(57.0–68.5)	286	76.2	(68.1–82.8)
Total	370	60.2	(53.3–66.8)	837	62.2	(56.3–67.7)	315	77.7	(70.4–83.6)
*Caregiver believes 2YL vaccination supported by community*									
No	71	59.7	(47.1–71.1)	95	57.0	(42.3–70.5)	18	77.4	(41.6–94.3)
Yes	299	60.4	(53.2–67.2)	742	62.8	(57.1–68.2)	297	77.7	(69.9–84.0)
Total	370	60.2	(53.3–66.8)	837	62.2	(56.3–67.7)	315	77.7	(70.4–83.6)
*Most people caregiver knows bring child in for 2YL vaccination*									
No	42	53.9	(38.3–68.8)	118	55.3	(42.4–67.6)	21	86.0	(66.1–95.1)
Yes	328	61.1	(53.5–68.2)	719	63.3	(57.4–68.8)	294	77.0	(69.3–83.2)
Total	370	60.2	(53.3–66.8)	837	62.2	(56.3–67.7)	315	77.7	(70.4–83.6)
*Caregiver believes it is important for child to get missing doses*									
No	24	51.2	(27.7–74.1)	**47**	**46.0**	**(29.6–63.3)**	12	89.6	(62.2–97.8)
Yes	346	60.8	(53.4–67.7)	**790**	**63.2**	**(57.7–68.4)**	303	77.2	(69.6–83.3)
Total	370	60.2	(53.3–66.8)	**837**	**62.2**	**(56.3–67.7)**	315	77.7	(70.4–83.6)
*Caregiver informed of 18 mos. visit by healthcare provider*									
No	178	56.9	(47.9–65.4)	**469**	**55.6**	**(48.8–62.2)**	**165**	**70.5**	**(60.7–78.7)**
Yes	192	62.8	(52.4–72.1)	**368**	**71.1**	**(63.8–77.5)**	**150**	**85.4**	**(79.0–90.1)**
Total	370	60.2	(53.3–66.8)	**837**	**62.2**	**(56.3–67.7)**	**315**	**77.7**	**(70.4–83.6)**
**B**									
		** *GAR* **			** *NR* **			** *VR* **	
	** *n* **	**%**	**95% CI**	** *n* **	**%**	**95% CI**	** *n* **	**%**	**95% CI**
*Child’s age group (months)*									
18–23	146	61.0	(54.3–67.3)	**302**	**58.7**	**(50.1–66.7)**	**115**	**69.2**	**(59.6–77.3)**
24–35	224	71.6	(64.4–77.8)	**535**	**76.4**	**(69.5–82.1)**	**200**	**91.2**	**(83.2–95.6)**
Total	370	67.4	(62.0–72.3)	**837**	**69.8**	**(63.2–75.8)**	**315**	**82.8**	**(76.0–87.9)**
*Child’s sex*									
Female	184	69.4	(61.9–76.0)	**396**	**74.2**	**(68.4–79.2)**	177	81.3	(70.3–88.9)
Male	186	65.3	(56.9–72.8)	**441**	**66.0**	**(56.6–74.2)**	138	84.8	(76.9–90.3)
Total	370	67.4	(62.0–72.3)	**837**	**69.8**	**(63.2–75.8)**	315	82.8	(76.0–87.9)
*Birth order*									
First	122	73.0	(64.2–80.3)	182	71.4	(61.8–79.4)	**101**	**86.1**	**(76.8–92.0)**
Second	100	66.7	(54.8–76.8)	175	68.1	(57.3–77.3)	**62**	**90.1**	**(81.2–95.1)**
Third or more	148	62.5	(54.2–70.0)	480	69.8	(63.2–75.8)	**152**	**76.8**	**(66.2–84.8)**
Total	370	67.4	(62.0–72.3)	837	69.8	(63.2–75.8)	**315**	**82.8**	**(76.0–87.9)**
*Maternal age (years)*									
<25	42	61.3	(44.6–75.8)	**108**	**57.4**	**(46.8–67.4)**	71	85.1	(74.0–92.0)
25–34	184	72.1	(63.0–79.6)	**215**	**72.6**	**(65.5–78.8)**	134	82.1	(72.7–88.8)
≥35	144	63.0	(52.9–72.1)	**514**	**71.4**	**(63.8–78.0)**	110	81.9	(71.9–88.9)
Total	370	67.4	(62.0–72.3)	**837**	**69.8**	**(63.2–75.8)**	315	82.8	(76.0–87.9)
*Maternal education*									
Never attended school	25	63.8	(45.2–79.1)	592	72.3	(65.9–78.0)	52	79.9	(65.6–89.3)
Primary	37	59.4	(41.9–74.8)	70	58.0	(37.2–76.4)	61	78.6	(62.8–88.8)
Secondary or higher	308	68.3	(62.3–73.8)	173	66.8	(56.6–75.6)	200	84.4	(76.3–90.1)
Total	370	67.4	(62.0–72.3)	835	69.8	(63.1–75.7)	313	82.7	(75.9–87.8)
*Paternal education*									
Never attended school	12	73.8	(56.5–86.0)	582	71.9	(65.1–77.8)	38	75.5	(63.8–84.4)
Primary	12	60.2	(28.2–85.3)	42	68.8	(44.7–85.8)	29	84.9	(64.5–94.6)
Secondary or higher	319	68.8	(63.3–73.8)	195	65.6	(55.1–74.9)	224	85.1	(76.2–91.1)
Total	343	68.6	(63.5–73.2)	819	70.1	(63.5–76.0)	291	84.0	(76.6–89.4)
*Maternal religion*									
Christian	314	67.9	(61.7–73.6)	**178**	**81.8**	**(71.2–89.0)**	272	83.7	(76.4–89.0)
Muslim	55	62.2	(49.6–73.3)	**590**	**65.8**	**(58.2–72.6)**	23	75.4	(55.3–88.3)
Traditionalist	0	–	–	**55**	**89.2**	**(75.6–95.7)**	6	76.7	(32.2–95.8)
None/Don’t know	1	100.0	–	**14**	**59.9**	**(30.4–83.6)**	14	77.7	(45.4–93.6)
Total	370	67.4	(62.0–72.3)	**837**	**69.8**	**(63.2–75.8)**	315	82.8	(76.0–87.9)
*Maternal marital status*									
Single/Divorced/Separated/Widowed	48	64.2	(48.9–77.2)	16	69.3	(52.7–82.0)	39	85.7	(65.7–95.0)
Married/Cohabitating	322	67.8	(61.9–73.3)	821	69.8	(63.1–75.9)	276	82.4	(75.4–87.7)
Total	370	67.4	(62.0–72.3)	837	69.8	(63.2–75.8)	315	82.8	(76.0–87.9)
*Settlement type*									
Urban	324	66.8	(61.1–72.0)	**224**	**57.4**	**(49.2–65.1)**	93	77.7	(64.7–86.9)
Rural	46	72.9	(63.2–80.8)	**613**	**74.9**	**(67.0–81.4)**	222	84.8	(76.1–90.7)
Total	370	67.4	(62.0–72.3)	**837**	**69.8**	**(63.2–75.8)**	315	82.8	(76.0–87.9)
*Attends school/daycare offering routine immunization*									
No formal school/daycare attendance	**179**	**62.4**	**(55.2–69.0)**	737	69.7	(62.4–76.2)	**221**	**79.1**	**(70.8–85.4)**
School/daycare does not offer RI	**90**	**67.2**	**(56.5–76.4)**	81	67.5	(56.2–77.0)	**48**	**99.4**	**(95.8–99.9)**
School/daycare offers RI	**101**	**75.9**	**(67.6–82.7)**	19	85.7	(65.3–95.0)	**46**	**85.2**	**(68.9–93.7)**
Total	**370**	**67.4**	**(62.0–72.3)**	837	69.8	(63.2–75.8)	**315**	**82.8**	**(76.0–87.9)**
*Timely receipt of all 6–week infant vaccines*									
At least 1 recommended dose delayed	40	86.0	(72.3–93.5)	282	72.0	(63.6–79.2)	53	88.3	(72.9–95.5)
All recommended doses timely	251	78.9	(74.0–83.1)	458	71.6	(64.5–77.8)	230	85.7	(80.2–89.9)
Total	291	79.9	(75.1–83.9)	740	71.8	(65.1–77.6)	283	86.1	(81.1–90.0)
*Timely receipt of all 14–week infant vaccines*									
At least 1 recommended dose delayed	94	80.0	(71.6–86.4)	511	73.5	(66.3–79.6)	107	85.9	(76.2–92.1)
All recommended doses timely	201	82.0	(76.6–86.3)	230	74.4	(66.5–81.0)	170	84.9	(77.6–90.1)
Total	295	81.3	(77.1–85.0)	741	73.8	(67.7–79.1)	277	85.3	(79.4–89.7)
*Timely receipt of all 9 mos. infant vaccines*									
At least 1 recommended dose delayed	87	79.1	(66.0–88.1)	381	74.8	(68.2–80.5)	**123**	**81.4**	**(71.1–88.6)**
All recommended doses timely	198	85.5	(80.5–89.5)	282	80.6	(73.7–86.0)	**133**	**91.5**	**(86.4–94.8)**
Total	285	83.7	(79.4–87.1)	663	77.4	(72.1–81.9)	**256**	**87.0**	**(81.0–91.2)**
*Fully immunized child by 12 mos.*									
No	**54**	**65.4**	**(50.1–78.1)**	**349**	**62.0**	**(54.3–69.1)**	89	81.0	(67.9–89.5)
Yes	**261**	**84.0**	**(80.1–87.3)**	**458**	**79.9**	**(73.8–84.9)**	216	86.9	(81.3–91.0)
Total	**315**	**81.0**	**(76.7–84.7)**	**807**	**72.2**	**(65.8–77.9)**	305	85.2	(79.7–89.4)
*18 mos. well–child visit attendance*									
No	**48**	**14.8**	**(5.9–32.4)**	**125**	**31.3**	**(19.7–45.9)**	**32**	**56.3**	**(36.2–74.5)**
Yes	**322**	**75.8**	**(69.9–80.9)**	**712**	**77.6**	**(72.6–82.0)**	**283**	**85.2**	**(78.1–90.4)**
Total	**370**	**67.4**	**(62.0–72.3)**	**837**	**69.8**	**(63.2–75.8)**	**315**	**82.8**	**(76.0–87.9)**
*Caregiver aware of need for 18 mos. visit before 2nd birthday*									
No	163	64.4	(54.2–73.5)	**452**	**64.2**	**(54.5–72.8)**	**151**	**74.4**	**(64.6–82.3)**
Yes	207	69.2	(59.3–77.6)	**385**	**76.7**	**(71.3–81.5)**	**164**	**90.1**	**(84.7–93.8)**
Total	370	67.4	(62.0–72.3)	**837**	**69.8**	**(63.2–75.8)**	**315**	**82.8**	**(76.0–87.9)**
*Caregiver knows MCV dose should be given during 2YL*									
No	**153**	**60.0**	**(51.6–67.9)**	**284**	**62.6**	**(52.4–71.8)**	107	81.4	(72.4–88.0)
Yes	**217**	**73.1**	**(65.9–79.3)**	**553**	**73.9**	**(67.9–79.1)**	208	83.6	(75.9–89.2)
Total	**370**	**67.4**	**(62.0–72.3)**	**837**	**69.8**	**(63.2–75.8)**	315	82.8	(76.0–87.9)
*Caregiver received phone reminder for 18 mos. visit*									
No access to phone	15	39.9	(16.4–69.2)	**233**	**78.2**	**(69.7–84.8)**	45	72.5	(57.7–83.6)
Access to phone but no reminder received	313	67.0	(59.9–73.4)	**523**	**65.9**	**(58.5–72.6)**	223	83.6	(76.1–89.1)
Phone reminder received	42	78.2	(59.3–89.9)	**81**	**77.3**	**(65.1–86.1)**	47	88.1	(74.0–95.1)
Total	370	67.4	(62.0–72.3)	**837**	**69.8**	**(63.2–75.8)**	315	82.8	(76.0–87.9)
*Caregiver believes they get enough 2YL vaccination info from provider*									
No	42	68.2	(54.6–79.3)	**114**	**58.0**	**(45.1–70.0)**	24	78.1	(53.9–91.5)
Yes	328	67.2	(61.1–72.8)	**723**	**71.5**	**(65.0–77.2)**	291	83.3	(75.8–88.8)
Total	370	67.4	(62.0–72.3)	**837**	**69.8**	**(63.2–75.8)**	315	82.8	(76.0–87.9)
*Caregiver believes 2YL vaccine is important for child*									
No	11	68.1	(43.4–85.5)	**31**	**46.3**	**(28.6–65.1)**	6	91.4	(54.0–99.0)
Yes	359	67.3	(62.0–72.3)	**806**	**71.0**	**(64.8–76.5)**	309	82.6	(75.8–87.8)
Total	370	67.4	(62.0–72.3)	**837**	**69.8**	**(63.2–75.8)**	315	82.8	(76.0–87.9)
*Caregiver believes 2YL vaccination supported by spouse*									
No	22	62.3	(39.5–80.7)	**29**	**40.5**	**(19.9–65.1)**	21	85.4	(67.3–94.3)
Yes	348	67.6	(62.2–72.6)	**808**	**71.0**	**(64.8–76.5)**	294	82.5	(75.3–88.0)
Total	370	67.4	(62.0–72.3)	**837**	**69.8**	**(63.2–75.8)**	315	82.8	(76.0–87.9)
*Caregiver believes 2YL vaccination supported by other family members*									
No	34	61.3	(45.6–74.9)	**53**	**46.2**	**(25.1–68.9)**	17	93.5	(75.3–98.5)
Yes	336	67.9	(62.7–72.8)	**784**	**71.5**	**(65.7–76.6)**	298	82.1	(75.2–87.3)
Total	370	67.4	(62.0–72.3)	**837**	**69.8**	**(63.2–75.8)**	315	82.8	(76.0–87.9)
*Caregiver believes 2YL vaccination supported by friends*									
No	66	68.7	(56.8–78.5)	71	63.1	(47.2–76.5)	29	90.0	(64.8–97.8)
Yes	304	67.1	(60.6–72.9)	766	70.5	(64.1–76.3)	286	81.9	(74.8–87.4)
Total	370	67.4	(62.0–72.3)	837	69.8	(63.2–75.8)	315	82.8	(76.0–87.9)
*Caregiver believes 2YL vaccination supported by community*									
No	71	72.9	(63.2–80.8)	95	62.2	(45.9–76.2)	18	81.3	(44.3–96.0)
Yes	299	66.0	(59.4–72.0)	742	70.7	(64.5–76.3)	297	82.8	(76.0–88.0)
Total	370	67.4	(62.0–72.3)	837	69.8	(63.2–75.8)	315	82.8	(76.0–87.9)
*Most people caregiver knows bring child in for 2YL vaccination*									
No	42	69.6	(51.6–83.1)	**118**	**58.0**	**(45.0–70.0)**	21	86.0	(66.1–95.1)
Yes	328	67.0	(61.1–72.5)	**719**	**71.6**	**(65.3–77.2)**	294	82.5	(75.5–87.8)
Total	370	67.4	(62.0–72.3)	**837**	**69.8**	**(63.2–75.8)**	315	82.8	(76.0–87.9)
*Caregiver believes it is important for child to get missing doses*									
No	24	62.7	(38.9–81.7)	**47**	**47.2**	**(30.1–64.9)**	12	100.0	–
Yes	346	67.6	(62.0–72.7)	**790**	**71.3**	**(65.2–76.6)**	303	82.0	(75.2–87.2)
Total	370	67.4	(62.0–72.3)	**837**	**69.8**	**(63.2–75.8)**	315	82.8	(76.0–87.9)
*Caregiver informed of 18 mos. visit by healthcare provider*									
No	178	64.8	(55.6–73.0)	**469**	**64.7**	**(55.5–72.9)**	**165**	**75.9**	**(66.1–83.5)**
Yes	192	69.3	(59.3–77.8)	**368**	**76.8**	**(71.1–81.6)**	**150**	**90.1**	**(84.1–94.0)**
Total	370	67.4	(62.0–72.3)	**837**	**69.8**	**(63.2–75.8)**	**315**	**82.8**	**(76.0–87.9)**

* Sample sizes are unweighted, percentages and 95% CIs are weighted. MACV, meningococcal serogroup A conjugate vaccine. MCV2, second dose of measles–containing vaccine. Bold values denote statistical significance of the design-based Chi-square test at the *p* < 0.05 level.

**Table 4 vaccines-11-01515-t004:** (**A**): Univariable and multivariable analysis of predictors of MACV receipt among children aged 18–35 months in Greater Accra Region (GAR), Northern Region (NR), and Volta Region (VR), Ghana, 2020. (**B**): Univariable and multivariable analysis of predictors of MCV2 receipt among children aged 18–35 months in Greater Accra Region (GAR), Northern Region (NR), and Volta Region (VR), Ghana, 2020.

A												
	GAR (*N* = 370)				NR (*N* = 870)				VR (*N* = 315)			
	Unadjusted		Adjusted		Unadjusted		Adjusted		Unadjusted		Adjusted	
	**OR**	**95% CI**	**OR**	**95% CI**	**OR**	**95% CI**	**OR**	**95% CI**	**OR**	**95% CI**	**OR**	**95% CI**
**Child’s age (months)**												
18–23	REF		REF		REF		REF		REF		REF	
24–35	2.2	(1.4–3.4) ***	4.0	(2.2–7.0) ***	2.4	(1.9–3.2) ***	3.0	(2.2–4.2) ***	6.6	(3.0–14.3) ***	7.2	(3.3–15.7) ***
**Child’s sex**												
Female	REF		REF		REF		REF		REF		REF	
Male	0.9	(0.5–1.6)	REF7	(0.5–1.9)	0.7	(0.5–0.9)	0.7	(0.5–1.0)	1.8	(0.8–3.9)	1.6	(0.6–3.9)
**Birth order**												
First	REF		REF		REF		REF		REF		REF	
Second	0.8	(0.3–1.6)	0.7	(0.2–2.0)	0.8	(0.5–1.2)	0.8	(0.4–1.4)	0.9	(0.3–2.8)	1.2	(0.4–3.4)
Third or more	0.7	(0.4–1.3)	0.5	(0.2–1.2)	0.8	(0.5–1.2)	0.5	(0.3–1.0)	0.5	(0.2–0.8) **	0.3	(0.2–0.7) **
**Maternal education**												
Never attended school	REF		REF		REF		REF		REF		REF	
Primary	0.6	(0.2–2.1)	1.0	(0.3–3.6)	0.5	(0.2–0.9)	0.5	(0.2–1.1)	1.1	(0.4–2.7)	2.0	(0.6–6.5)
Secondary or higher	0.8	(0.3–1.9)	1.1	(0.5–2.4)	0.9	(0.6–1.2)	0.9	(0.6–1.4)	1.3	(0.6–2.4)	1.4	(0.5–4.0)
**Maternal age (years)**												
<25	REF		REF		REF		REF		REF		REF	
25–34	1.7	(0.7–3.8)	1.8	(0.7–4.3)	2.1	(1.2–3.3) **	2.2	(1.4–3.4)	0.8	(0.3–1.8)	1.0	(0.4–2.3)
≥35	1.4	(0.6–3.0)	2.7	(1.1–6.4) *	1.6	(1.1–2.2) *	1.8	(1.2–2.7) **	0.7	(0.3–1.5)	1.1	(0.4–2.9)
**Maternal religion**												
Other	REF		REF		REF		REF		REF		REF	
Christian	1.2	(0.6–2.5)	1.0	(0.4–2.8)	2.2	(1.3–3.7) **	1.9	(1.1–3.5) *	1.3	(0.6–3.0)	0.8	(0.2–2.8)
**Settlement type**												
Urban	REF		REF		REF		REF		REF		REF	
Rural	1.7	(0.9–3.2)	2.7	(1.2–5.9) **	1.8	(1.1–2.8)	1.9	(1.2–3.0) ***	1.0	(0.4–2.4)	1.0	(0.4–2.6)
**Fully immunized child by 12 mos. of age**												
No	REF		REF		REF		REF		—	—	—	—
Yes	2.0	(1.1–4.0) *	3.1	(1.4–7.1) **	2.1	(1.6–2.8) ***	2.4	(1.7–3.4) ***	—	—	—	—
**Caregiver knows MACV dose should be given during 2YL**												
No	—	—	—	—	REF		—	—	REF		—	—
Yes	—	—	—	—	1.7	(1.1–2.5) *	—	—	1.8	(0.8–3.9)	—	—
**Caregiver aware of need for 18 mos. visit before 2nd birthday**												
No	—	—	—	—	REF		REF		REF		REF	
Yes	—	—	—	—	2.0	(1.4–2.8) ***	1.9	(1.3–2.6) ***	2.8	(1.6–4.6) ***	2.8	(1.7–4.5) ***
**Child attends school/daycare offering routine immunization**												
No	REF		—	—	—	—	—	—	—	—	—	—
Yes	2.2	(1.4–3.4) **	—	—	—	—	—	—	—	—	—	—
**Caregiver believes 2YL vaccine is important for their child**												
No	—	—	—	—	REF		REF		—	—	—	—
Yes	—	—	—	—	2.2	(1.1–4.2) *	1.9	(1.1–3.3) *	—	—	—	—
**Caregiver believes it is important for child to get missing doses**												
No	—	—	—	—	REF		—	—	—	—	—	—
Yes	—	—	—	—	2.0	(1.0–3.9) *	—	—	—	—	—	—
**Caregiver agrees that 2YL vaccination is supported by spouse**												
No	—	—	—	—	REF		—	—	—	—	—	—
Yes	—	—	—	—	2.5	(0.9–6.8)	—	—	—	—	—	—
**Caregiver believes that 2YL vaccination supported by other family members**												
No	—	—	—	—	REF		—	—	—	—	—	—
Yes	—	—	—	—	2.6	(1.1–5.7) *	—	—	—	—	—	—
**Most people caregiver knows bring child in for 2YL vaccination**												
No	—	—	—	—	REF		—	—	—	—	—	—
Yes	—	—	—	—	1.3	(0.8–2.2)	—	—	—	—	—	—
**Caregiver believes healthcare provider provides enough information on 2YL vaccination**												
No	—	—	—	—	REF		—	—	—	—	—	—
Yes	—	—	—	—	1.4	(0.8–2.3)	—	—	—	—	—	—
**Maternal occupation**												
Unemployed/Housewife	—	—	—	—	REF		—	—	—	—	—	—
Farmer/Laborer/Fisherwoman	—	—	—	—	0.9	(0.5–1.6)	—	—	—	—	—	—
Civil servant	—	—	—	—	0.7	(0.2–2.5)	—	—	—	—	—	—
Other occupation	—	—	—	—	0.5	(0.3–1.1)	—	—	—	—	—	—
Trader/Merchant	—	—	—	—	0.5	(0.2–1.1)	—	—	—	—	—	—
**Caregiver received phone reminder for 18 mos. visit**												
No	—	—	—	—	REF		—	—	—	—	—	—
Yes	—	—	—	—	1.4	(0.8–2.6)	—	—	—	—	—	—
**B**												
	**GAR (*N* = 370)**				**NR (*N* = 870)**				**VR (*N* = 315)**			
	**Unadjusted**		**Adjusted**		**Unadjusted**		**Adjusted**		**Unadjusted**		**Adjusted**	
	**OR**	**95% CI**	**OR**	**95% CI**	**OR**	**95% CI**	**OR**	**95% CI**	**OR**	**95% CI**	**OR**	**95% CI**
**Child’s age (months)**												
18–23	REF		REF		REF		REF		REF		REF	
24–35	1.6	(1.0–2.4) *	3.2	(1.7–6.2) ***	2.2	(1.6–3.0) ***	2.9	(2.1–4.1) ***	4.6	(2.1–9.9) ***	4.6	(2.1–10.5) ***
**Child’s sex**												
Female	REF		REF		REF		REF		REF		REF	
Male	0.8	(0.4–1.3)	0.9	(0.5–1.9)	0.6	(0.4–0.9) *	0.7	(0.4–1.1)	1.2	(0.5–3.0)	1.2	(0.5–2.7)
**Birth order**												
First	REF		REF		REF		REF		REF		REF	
Second	0.7	(0.3–1.4)	0.6	(0.2–2.0)	0.8	(0.5–1.4)	0.8	(0.4–1.4)	1.4	(0.5–4.2)	1.5	(0.4–4.7)
Third or more	0.6	(0.3–1.1)	0.3	(0.1–0.8) *	0.9	(0.6–1.4)	0.6	(0.3–1.1)	0.5	(0.3–0.9) *	0.4	(0.2–0.8) **
**Maternal education**												
Never attended school	REF		REF		REF		REF		REF		REF	
Primary	0.8	(0.2–2.5)	1.4	(0.4–4.9)	0.5	(0.2–1.2)	0.7	(0.3–1.9)	0.9	(0.2–2.9)	1.2	(0.4–3.8)
Secondary or higher	1.2	(0.5–2.8)	1.9	(0.8–4.2)	0.7	(0.4–1.2)	1.0	(0.6–1.6)	1.3	(0.6–3.0)	1.3	(0.5–3.4)
**Maternal age (years)**												
<25	REF		REF		REF		REF		REF		REF	
25–34	1.6	(0.6–3.9)	1.8	(0.4–8.2)	2.0	(1.2–3.2) **	2.1	(1.3–3.4) **	0.8	(0.3–1.7	1.0	(0.4–2.1)
≥35	1.0	(0.5–2.0)	2.2	(0.7–7.2)	1.9	(1.2–2.7) **	2.2	(1.4–3.3) ***	0.7	(0.2–2.2)	1.1	(0.4–2.8)
**Maternal religion**												
Other	REF		REF		REF		REF		REF		REF	
Christian	1.2	(0.6–2.2)	1.1	(0.4–2.4)	2.2	(1.1–4.2) *	1.9	(0.9–3.7)	1.6	(0.7–3.5)	1.1	(0.4–2.6)
**Settlement type**												
Urban	REF		REF		REF		REF		REF		REF	
Rural	1.3	(0.7–2.2)	1.8	(0.8–4.0)	2.2	(1.3–3.7) **	2.6	(1.5–4.3) ***	1.5	(0.6–3.8)	1.7	(0.6–4.4)
**Fully immunized child by 12 mos. of age**												
No	REF		REF		REF		REF			—	—	—
Yes	2.7	(1.4–5.5) **	3.8	(1.6–8.5) **	2.4	(1.7–3.3) ***	2.9	(1.9–4.3) ***		—	—	—
**Caregiver knows MCV dose should be given during 2YL**												
No	REF		REF		REF		—	—		—	—	—
Yes	1.8	(1.1–2.9) *	2.4	(1.2–4.9) *	1.7	(1.1–2.5) **	—	—		—	—	—
**Caregiver aware of need for 18 mos. visit before 2nd birthday**												
No	—	—	—	—	REF		REF		REF		REF	
Yes	—	—	—	—	1.8	(1.2–2.7) ***	1.7	(1.1–2.5) **	3.1	(1.9–5.3) ***	2.7	(1.7–4.3) ***
**Child attends school/daycare offering routine immunization**												
No	REF		—	—	REF		—	—	—	—	—	—
Yes	1.8	(1.1–2.9) ***	—	—	2.6	(0.7–9.1)	—	—	—	—	—	—
**Caregiver believes 2YL vaccine is important for their child**												
No	—	—	—	—	REF		REF		—	—	—	—
Yes	—	—	—	—	2.8	(1.3–5.9) **	2.8	(1.4–5.6) **	—	—	—	—
**Caregiver believes it is important for child to get missing doses**												
No	—	—	—	—	REF		—	—	—	—	—	—
Yes	—	—	—	—	2.7	(1.4–5.3) **	—	—	—	—	—	—
**Caregiver agrees that 2YL vaccination is supported by spouse**												
No	—	—	—	—	REF		—	—	—	—	—	—
Yes	—	—	—	—	3.6	(1.3–9.6) *	—	—	—	—	—	—
**Caregiver believes that 2YL vaccination supported by other family members**												
No	—	—	—	—	REF		—	—	REF		—	—
Yes	—	—	—	—	2.9	(1.2–6.9) *	—	—	0.3	(0.0–1.5)	—	—
**Most people caregiver knows bring child in for 2YL vaccination**												
No	—	—	—	—	REF		—	—	—	—	—	—
Yes	—	—	—	—	1.8	(1.1–3.0) *	—	—	—	—	—	—
**Caregiver believes healthcare provider provides enough information on 2YL vaccination**												
No	—	—	—	—	REF		—	—	—	—	—	—
Yes	—	—	—	—	1.8	(1.1–2.9) *	—	—	—	—	—	—
**Paternal education**												
Never attended school	—	—	—	—	REF		—	—	REF		—	—
Primary	—	—	—	—	0.8	(0.3–2.1)	—	—	1.8	(0.4–7.4)	—	—
Secondary or higher	—	—	—	—	0.7	(0.4–1.2)	—	—	1.8	(0.8–4.1)	—	—

** p*-value < 0.05; ** *p*-value < 0.01; *** *p*-value < 0.001.

## Data Availability

The datasets used and/or analyzed during the current project may be available upon reasonable request.

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
