# Peer review of "Predictors for Uptake of Vaccines Offered during the Second Year of Life: Second Dose of Measles-Containing Vaccine and Meningococcal Serogroup A-Containing Vaccine, Ghana, 2020"

_vaccines, 2023, doi:10.3390/vaccines11101515_

Round 1

Reviewer 1 Report

This is an interesting and well-written paper describing the factors influencing immunisations during the second year of life in Ghana. The methodology is robust, and results are presented in a clear manner. The results are not highly surprising, and therefore in line with previous studies of similar nature in Ghana and comparable countries. The authors acknowledge this and reference relevant previous studies. 

Minor issues to be addressed: 1) Table 1 should mention number of study subjects; 2) Table 4A and 4B contains several blank fields; 3) The conclusion is somewhat vague and fails to clearly mention some of the important findings of the study, e.g. that children with older siblings are less likely to be fully year 2 vaccinated 

Reviewer 2 Report

The authors have addressed a very important problem in this manuscript. The authors have concluded that the enhancement in the uptake of infant immunization can be achieved by increasing community awareness and employing targeted strategies. For instance, implementing parental reminders for vaccination visits could potentially boost vaccination coverage for second year of life (2YL).

1. I would recommend to shorten the title, as it is quite long and not very clear.

2. Have authors consider the batch effect as the data is coming from three different regions. If yes, then what methodology the authors have used to remove the batch effect.

3. Have authors considered the regions as covariates, from which the samples belong.

4. While writing p-value in the text or on the tables, do not capitalize the "p" in that.

5. In Tables, what do authors want to signify with the blank cells. I would recommend to write "N/A" instead of leaving them blank.

6. The conclusion of the study is not well-written. I would recommend the authors to rewrite a strong conclusion.

7. I would like to suggest the authors to provide the flow-chart of their analysis representing the overall flow the work they have conducted.

8. The authors have shown the p-values in their analysis. However, I could not find the description of the tests they have applied to get the p-value signifying the difference.

9. Does the authors checked the normality of their data, to decide the parametric or non-parametric test for their variables.

10. The authors have mentioned the chi-square test in the tables, please mention the same in the text of the manuscript. Also provide the rationale behind their decision to choose the chi-square test.

1. Minor grammatical and punctuation errors are present in the manuscript. I would like to recommend the authors to read it thoroughly and rectify the errors.

Reviewer 3 Report

I read with interest the manuscript submitted to me for review. 

Apart the slighty excessive length, that I would suggest to the authors, if possible, to shorten, my considerations are as follows:

Abstract: presented in an understable form. The predictors for uptake of vaccines are correctly indicated.

Introduction: complete with a good coverage of the liberatore.

Mat & Meth: study design, population and variables are well identifiex. The statistical analysis is adequate.

Results: although the tables are very long, they are still understable.

Discussion and Conclusions are sound withthe study design.

References: adeguate. 

Reviewer 4 Report

Title:  Predictors for uptake of vaccines offered during the second year of life: Second dose of measles and rubella-containing vaccine and meningococcal serogroup A-containing vaccine, Ghana, 2020.

Manuscript ID: vaccines-2551085

The manuscript is very interest and highline the need to improve infant immunization, which could addressed new and different strategies to better vaccination coverage. I recommended that manuscript could be accepted with MINOR MODIFICATIONS.

-          Include a map with studied regions.

-          Compare results from Ghana with other countries in Africa or countries in developed.

-          The study was conducted during COVID-19 pandemic. Specify if no interference was observed in this regards.  

-          Review abbreviations.

Round 2

Reviewer 2 Report

The authors have addressed my queries satisfactorily.